# Progress in Research on the Gut Microflora of the Red Panda (*Ailurus fulgens*)

**DOI:** 10.3390/microorganisms12030478

**Published:** 2024-02-27

**Authors:** Xing Zhao, Zejun Zhang, Le Wang, Qian Zhang, Liwen Kang, Jia Wang, Juejie Long, Mingsheng Hong

**Affiliations:** 1Liziping Giant Panda’s Ecology and Conservation Observation and Research Station of Sichuan Province (Science and Technology Department of Sichuan Province), China West Normal University, Nanchong 637001, China; zukoxi98@163.com (X.Z.); zhangzejun66@163.com (Z.Z.); wangle_0806@163.com (L.W.); liwenkang96@163.com (L.K.); wj602602@163.com (J.W.); ljj13440088076@163.com (J.L.); 2Key Laboratory of Southwest China Wildlife Resources Conservation (Ministry of Education), China West Normal University, Nanchong 637002, China; 3Appraisal Center for Environment and Engineering, Ministry of Ecology and Environment, Beijing 100006, China; zhangqian@acee.org.cn

**Keywords:** red panda, gut microflora, next generation sequencing technology (NGST), captive individual, microbial diversity, protection

## Abstract

Animals can adapt to unique feeding habits through changes in the structure and function of the gut microflora. However, the gut microflora is strongly influenced by the evolutionary relationships between the host, nutritional intake, intake of microorganisms, etc. The red panda (*Ailurus fulgens*), an herbivorous carnivore, has adapted to consuming bamboo through seasonal foraging strategies and optimization of the composition and function of its gut microflora during long-term evolution. However, to date, studies of the gut bacteria of the red panda have mainly focused on the composition, diversity and function of the gut microflora of captive individuals. There are a lack of studies on how the wild red panda adapts to the consumption of bamboo, which is high in fibre and low in nutrients, through the gut microflora. This paper reviews the technology and methods used in published studies investigating the gut microflora of the red panda, as well as the composition, diversity and function of the identified microbes and the influencing factors. Furthermore, this paper suggests future research directions regarding the methodology employed in analyzing the red panda gut microflora, the interplay between gut microflora and the health of the red panda, the red panda’s adaptation to its gut microflora, and the implications of these studies for the management and conservation of wild red pandas. The goal of this review is to provide a reference for the protection of wild red pandas from the perspective of the gut microflora.

## 1. Introduction

Microorganisms in the gut are intricately linked to the host’s physiology, immunity, development and metabolism [1,2,3]. Many factors, such as the living environment and the host’s genetics, diet and microorganisms acquired at birth, have significant impacts on the composition, diversity and functions of the gut microflora [4,5,6].

Previous studies demonstrated significant differences in the composition, diversity and functional characteristics of the gut microflora between groups of humans that consumed high-fat foods as compared to those who consumed high-carbohydrate foods [7,8]. A diet that is high in carbohydrates can have a multiplying effect on the gut microbial diversity of carnivores, omnivores and herbivores [9]. Special dietary habits may lead to the overgrowth of some intestinal strains, resulting in changes in the intestinal pH and metabolism pathways, which may lead to the emergence of pathogenic microorganisms [10].

For wild animals, the type and nutritional composition of the food consumed, and the host’s evolutionary relationship with other species, have significant impacts on the composition and function of the gut microbiome [9,11,12,13]. The capacity of wild animals to utilize nutrients will change with changes in the gut microflora [13,14,15]. The unique eating habits of an animal result from the animal’s adaptation to its environment over a long period of time [16,17].

Animals adapt to changes in their feeding habits through specific gut microbes. Carnivores such as Felidae and Canidae primarily harbour microorganisms related to the digestion of high-purine and high-fat foods, while omnivores such as Procyonidae and Mustelidae and the herbivorous giant and red pandas harbour high proportions of microorganisms that degrade cellulose and hemicellulose [13,18]. In the case of herbivores, a series of behavioural and physiological mechanisms have developed during the process of co-evolution in order to adapt to a high-fibre diet. For example, the rumen of herbivores can efficiently digest high-fibre foods through fermentation [19]. Because of its lack of a complex stomach or cecum, there has been extensive research on the specific diet and gut microflora of the giant panda.

However, compared with the large number of studies on the gut microflora of the giant panda, there are few studies on the gut microflora of the red panda. With the widespread application of next-generation sequencing technology (NGST), in-depth studies on the gut microflora of the red panda have emerged. The main purpose of this paper was to review the microbial composition, diversity and functions of the red panda’s gut microflora and the research methods used in the available studies. Then, future research directions in this field are proposed. The goal of this review is to provide a reference for the protection of wild red pandas from the perspective of the gut microflora.

## 2. Methodology Used in Research on the Gut Microflora of the Red Panda

Early studies used traditional cultivation techniques and denaturing gradient gel electrophoresis (DGGE) to identify and classify the intestinal microbiota of the red panda. The former method employs selective media for pure cultivation and direct observation of microorganisms. Classification and identification were carried out based on the morphological, physiological and biochemical characteristics of the microorganisms [20,21,22,23,24,25]. With the gradual popularization of primer amplification techniques, Li et al. (2017) analysed the bacterial diversity in the gastrointestinal tract of deceased red pandas using DGGE targeting the V3 hypervariable region of the bacterial 16S rRNA gene. The study revealed a distinct pattern of high to low bacterial diversity along the digestive tract, progressing from the stomach, duodenum, jejunum, ileum and colon to the rectum [26].

The rapid development of high-throughput sequencing technologies (HTST) has facilitated in-depth research on the gut microbiota of the red panda. Pyrosequencing technology quickly gained favour among researchers studying the gut microbiota because of its higher sequencing accuracy and depth. Crowe et al. (2013) employed pyrosequencing technology using the Roche/454 GS Junior platform to sequence the gut bacteria of the red panda, revealing that the structure of the red panda gut microbiome was significantly influenced by its age, gender and geographic location [27]. Using the same approach, Williams et al. (2014) discovered that the cellulose-degrading microbial community in the red panda differs from that of the giant panda [28]. In another study, the Roche GS FLX Titanium platform was used to target the V1–V3 hypervariable regions of the bacterial 16S rRNA gene and found that the diversity, richness and evenness of the gut microbiota of wild red pandas were higher than that of captive individuals [29]. In a study by Huang et al. (2020), the high similarity in the gut microbiota of the red panda and giant panda was found to be driven by their similar diet, as compared to other herbivorous and carnivorous animals [30]. In another study, the gut microbiota of the red panda was found to be distinct from those of the giant panda and the black bear [31]. Using the Ion PGM platform, McKenney et al. (2018) amplified the V2, V3, V4, V6, V8, and V9 hypervariable regions of the bacterial 16S rRNA gene and discovered substantial overlap in low-abundance gut microbes between captive red pandas and giant pandas [32].

In recent years, the use of the Illumina sequencing platform has become widespread. Williams et al. (2018) used the Illumina MiSeq platform (Illumina, San Diego, CA, USA) to amplify the V3–V4 regions of the bacterial 16S rRNA gene and found that weaning-induced dietary changes in captive red pandas affected the species composition and abundance of the gut microbiota [33]. Using the Illumina/Miseq PE300 platform, the gut microbiota diversity of the highly variable regions of the bacterial 16S rRNA genes V1–V3 and V3–V4 were found to differ significantly between seasons [34,35]. These differences, such as number of species and structural diversity of bacteria, were strongly influenced by dietary change, growth and development [36]. In another study, Zeng et al. (2018) used the Illumina HiSeq2500 platform to sequence the V4 hypervariable region of the bacterial 16S rRNA gene and found lower bacterial diversity in the stomach and duodenum of the red panda [37].

Metagenomic shotgun sequencing, renowned for its exceptionally low mismatch rate and ultra-high sequencing depth, was considered the pinnacle of gut microbial species identification techniques. Currently, metagenomics is widely employed to study gut microbes that have carbohydrate degradation functions in the red panda. Building on this approach and using an Illumina/HiSeq 2000, Huang et al. (2020) revealed that the symbiotic gut microbiota of the red panda contained high amounts of microbes with starch and sucrose metabolism and vitamin B12 biosynthesis functions [30]. Zhu et al. (2018) utilised metagenomic methods on the Illumina HiSeq2500 sequencing platform and discovered that microbes in the gut of the red panda carried genes associated with cyanide degradation [18]. Moreover, through metagenomics, we identified fundamental patterns in metabolic pathways such as the cellulase (EC3.2.1.4), 1,4-β-xylosidase (EC3.2.1.37) and β-glucosidase (EC3.2.1.21) pathways that responded to seasonal and dietary changes [35].

## 3. Composition of the Red Panda Gut Microflora

The gastrointestinal tract in animals is a massive microbial ecosystem containing trillions of microbial cells. There was a high microflora density in the faeces; the microflora density in the colon was similar to that in the faeces and higher than in other gut segments [38]. Thus, almost all studies have focused on faecal microbes (Table 1). Because of the difficulty of collecting samples, research on the gut microbiome of the red panda has mainly been conducted on captive animals; only five studies have examined wild individuals (Table 1).

Although multiple types of microbes (bacteria, fungi, archaea, viruses and so on) exist in the gut, almost all in the red panda gut are prokaryotic bacterial microorganisms. Firmicutes play an important role in the pre-weaning, weaning, post-weaning and adulthood developmental stages [33]. It is the most dominant phylum in the guts of captive red panda individuals, followed by the phyla Proteobacteria, Bacteroidetes and Actinobacteria [28,29,31,32,33,37]. However, the distributions of Firmicutes, Proteobacteria and Bacteroidetes in the intestines of wild red panda individuals were more uniform [39]. A metagenomics study found a relative distribution of Proteobacteria in the guts of wild red panda individuals of up to 77% [18]. Excessive Proteobacteria caused inflammatory bowel disease and metabolic syndrome; variations in the levels of Proteobacteria were related to the animal’s diet, species and whether it was captive or wild [29,31,37,39].

Even under artificial rearing conditions, the gut microbiota of rescued wild red panda cubs retained a significant degree of wild characteristics. The relative abundances of *Gammaproteobacteria* and *Bacilli* gradually decrease with age in red panda cubs, while the abundance of Clostridia increased from two to five months of age (Figure 1) [36]. When red panda cubs began to lick bamboo leaves at five months of age, there was a significant increase in the abundance of *Clostridia* in the gut [36]. At the order level, Kong et al. (2014) reported that the relative distributions of *Lactobacillus* and *Clostridium* in the guts of captive red pandas were related to the age and geographic location of the individual [29]. The relative abundance of *Clostridium* was the highest, followed by *Lactobacillus* and *Enterococcus*. *Clostriaceae* were dominant in captive red pandas at the family level, while the distribution of *Clostriaceae* was more even in the intestinal tract of wild individuals. Using metagenomic sequencing, Zhu et al. (2018) found that *Pseudomonaceae* were dominant in the guts of wild red panda individuals [18].

At the genus level, Williams et al. (2013) reported that *Clostridium* was dominant in the guts of captive red pandas (up to 75%) [28]. Many species in the genus *Clostridium* have been shown to degrade cellulose and hemicellulose to oligosaccharides and monosaccharides [35,36,40]. Li et al. (2015) found that *Sarcina* was dominant in the guts of captive red pandas. In other studies, high distributions of *Nitrococcus*, *Filomicrobium* and *Croceibacter* have been observed in the faeces and gastrointestinal tracts of deceased captive red pandas [26,37]. However, the available amplicon sequencing studies did not classify most of the red panda gut microbiota at the genus level. Using metagenomic sequencing, Zhu et al. (2018) found that Pseudomonas had the highest relative distribution in the guts of wild red panda individuals; a large number of other bacterial genera were also identified [18].

## 4. Alpha Diversity of Red Panda Gut Microflora

Alpha diversity is a measure of the richness and diversity of biological communities within a plot or sample [41]. Various alpha diversity indices can be used to estimate the species abundance and diversity of the microbial community. These species richness indices include Observed species, the Chao1 estimator (Chao1), the ACE estimator (Ace) and so on. The Shannon index and the Simpson index are also often used to evaluate microbial community diversity in the animal gut [30,34,35].

Several studies have reported that the Simpson and Shannon indices of the intestinal bacterial microbial community of wild red pandas were significantly higher than those of captive individuals on the basis of pyrosequecing [29]. The Shannon index of captive red pandas was also found to be considerably lower than that of captive giant pandas and captive bamboo lemurs [32]. Further analyses have revealed that bamboo-eating giant pandas, red pandas and bamboo lemurs share a large number of low-abundance gut microbes [32]. The Shannon and Simpson indices of the intestinal bacteria of captive red pandas reach the highest levels during the weaning period (milk + leaf diet) [33]. Zeng et al. (2018) dissected a red panda that died in captivity and found that the numbers of bacterial species in the large intestine and faeces were lower than those in the small intestine and stomach, but the bacterial microbial diversity was the opposite [26,37]. Our research results revealed a higher Observed species and Shannon index in wild red panda faecal samples in winter and spring [35]. In another study, the Sobs index and Shannon index of gut bacterial microbes in rescued red panda cubs increased with age but were significantly lower than those of adult captive and wild red pandas (Figure 2) [36]. A comprehensive diet (dominant leaf with ancillary milk, apple, etc.) may be the reason for a higher gut microbial diversity in captive individuals than that in wild (leaf diet) and cub individuals (milk diet) [33,36].

## 5. Beta Diversity of the Red Panda Gut Microflora

Beta diversity reflects differences or distances in species diversity between samples or plots. Many indices can be used to measure beta diversity, such as the Jaccard index, Bray–Curtis, (un)weighted UniFrac distance and so on [42]. These indices can be estimated through principal coordinate analysis, nonmetric multidimensional scaling, etc. [33,43].

Weaning was an important dietary change, and Williams et al. (2018) found that the structure of the intestinal bacterial community of captive red pandas was most similar in the weaning stage and the post-weaning stage but was different from that of the pre-weaning stage and adulthood stage, according to the Bray–Curtis distance [33]. There was significant dissimilarity in the bacterial community structure of one captive dead red panda between the stomach, small intestine, large intestine and faeces [37]; the same research group also found some similarity in the bacterial community structure of this individual between adjacent intestinal segments [26].

In our study, there were significant differences in the inter-group Unweighted UniFrac distances of the gut bacterial microbes of two-month-old and five-month-old rescued red panda cubs, as well as between three-month-old and five-month-old cubs (*p* < 0.05, Kruskal–Wallis H test) (Figure 3A) [36]. However, analysis of the inter-group Weighted UniFrac distances showed significant differences only between four-month-old and five-month-old cubs (*p* < 0.05, Kruskal–Wallis H test) (Figure 3B) [36]. Overall, there was reduced similarity in the gut microbiota community structure of red panda cubs with increases in age (Figure 3A) [36].

Analysis of the Jaccard distance index and the Yue and Clayton (Theta YC) distance index showed that the intestinal bacterial community structure of captive red pandas was significantly different from that of the wild population [29]. Moreover, the community difference/distance between the intestinal microbial samples of captive red pandas was significantly smaller than that of the wild population, which may reflect the more heterogeneous living environment and food sources of wild red panda individuals than those of captive red panda individuals [29]. Seasonal differences in the gut bacterial community structure were also found in captive and wild red pandas [34,35].

In another study, the Bray–Curtis, ThetaYC, Weighted Unifrac and Morisita–Horn distance metrics revealed that the gut bacterial community structure of captive red pandas was significantly different from those of giant pandas and Asian black bears [31]. McKenney et al. (2018) also found that the captive red panda gut microbiota was significantly different from those of bamboo lemurs (*Hapalemur griseus*), giant pandas, lemurs (*Lemur catta*) and Asian black bears on the basis of the Unweighted UniFrac distance metric calculated using the Ion PGM sequencing platform [32]. However, analysis of the Bray–Curtis distance indicated that the bacterial community structure of wild red pandas was more similar to that of giant pandas but was dissimilar to those of carnivores, omnivores and David’s deer [18].

## 6. Functions of the Red Panda Gut Microflora

In order to adapt to exclusive feeding on bamboo, which is a low-fat, relatively poor-quality food, the red panda has undergone a series of adaptive changes in morphology, behaviour, ecology, genetics and intestinal microbes, including the development of pseudo-thumbs, well-developed zygomatic arches, modified sweet taste receptor gene TAS1R1 and the consumption of high-nutrient-content bamboo leaves and bamboo shoots [29,31,44,45]. The red panda has not only adapted to feeding on high-fibre, low-nutrient bamboo by optimizing the composition and function of its gut microflora but has also formed special gut microbes to adapt to the consumption of bamboo with high secondary metabolite contents [18,29,31]. For example, a phylogenetic tree constructed with the Neighbour-Joining algorithm based on Kimura 2-parameter distances revealed that 10 of the 50 OTUs with the highest relative distributions were associated with known cellulose-degrading bacteria [29]. Using macrogenomics, Zhu et al. (2018) found that wild red pandas had a high proportion of Pseudomonas (63%), considerably higher than that found in the intestines of giant red pandas (39%). The majority of microbial flora within the Pseudomonas genus exhibited the ability to degrade cyanide found in bamboo, and this capability was attributed to the presence of putative thiosulfate/3-mercaptopyruvate sulfurtransferases in the genomes of these bacteria [18].

In recent years, our research team has been dedicated to unravelling the mechanisms and determining the factors underlying the dynamic changes in the gut microbiota of the red panda. Our latest research has revealed that the seasonal dynamics of the gut microbiota of the red panda were primarily influenced by dietary variations across the different seasons. In the wild, during the leaf-eating phase and periods of mixed dietary intake characterized by high cellulose and hemicellulose contents, the functional abundances of cellulases, β-glucosidase and 1,4-β-xylosidases in the red panda’s gut microbiota significantly surpassesd those observed during the bamboo shoot consumption period [35]. This is similar to giant pandas, indicating that, akin to their larger counterparts, red pandas enhanced the concentrations of cellulases and hemicellulases within their gut microbiota to facilitate the breakdown of cellulose, hemicellulose and related compounds, thereby acquiring energy and nutritional resources.

Metagenomic analysis revealed that the symbiotic gut microbiota of the red panda possessed high levels of microbes with starch and sucrose metabolism and vitamin B12 biosynthesis functions, which was significantly different from the common ferret and the polar bear [30]. The dietary habits of giant pandas and red pandas overlapped substantially, leading to shared features in their respective gut microbial compositions. Notably, the structures of both species’ gut microbial communities were predominantly characterized by the phylum Proteobacteria. Furthermore, there was an abundance of genes related to cyanide detoxification in their gut flora, a trait believed to be closely associated with the adaptability of their digestive systems to dietary variations [18]. The sympatric Dian snub-nosed monkey (*Rhinopithecus roxellana*) demonstrated a gut microbial structure highly similar to that of the red panda, owing to a significant dietary overlap between the two species [46]. This shared bamboo-oriented dietary trait contributed to the prevalence of specific genes, including thiosulfate/3-mercaptopyruvate sulfurtransferase, nitrilase (TST) thiosulfate sulfurtransferase (glpE), cobalamin adenosyltransferase (EC 2.5.1.17) and nitrilase (EC 3.5.5.1) [46]. These genes flanked the expression of enzymes related to protein and cellulose degradation in bamboo shoots. The remarkable convergence in the gut microbial structures of the red panda and giant panda underscored the enduring impact of dietary influences over an extended period. This observation highlighted the remarkable plasticity of the mammalian gut flora. Overall, our past findings shed light on the intricate relationships between the diet, the gut microbial composition and the adaptive mechanisms employed by these species to thrive in their respective ecological niches.

## 7. Pathogenic Microflora in the Red Panda Gut

The wild red panda population was threatened by deforestation, habitat loss, poaching, livestock grazing and disease [45,47,48]. Mortality among captive red pandas was mainly due to infectious diseases, such as respiratory diseases, digestive system diseases, circulatory system diseases, urinary system infections and trauma [49]. *Klebsiella pneumoniae* was an important gram-negative opportunistic pathogen that caused pneumonia and respiratory damage in the red panda [50]. A previous report on the digestive diseases of the red panda indicated that viral infections such as canine parvovirus [51] and bacterial infections such as *Escherichia coli* and *Proteus mirabilis* were present in red panda individuals suffering from urinary system infection and circulatory system diseases, respectively [49]. In another study, the conditional pathogenic bacteria *Escherichia coli* and *Acinetobacter* were detected in the stomach and ileum, respectively, of the red panda. These two kinds of bacteria could lead to gastrointestinal diseases in red pandas with poor health [26].

## 8. Summary and Prospects

With HTST, researchers are looking more at the gut bacteria in red pandas. Sequencing platforms based on pyrosequencing technology have been widely employed, as have serial Illumina sequencing instruments. Targeting sequencing has also gradually appeared, from sequencing of the hypervariable regions of the bacterial 16S rRNA genes to the metagenome, and deeper functional issues are increasingly being clarified. Because of these improved biotechnologies, dominant bacterial microorganisms and their influencing factors (growth and development, seasons, diet, captivity vs. wild, comparisons with other species and so on) have come to light. However, other microbes, such as fungi, archaea and viruses in the red panda gut are yet to be fully studied.

At present, challenges persist, such as the overwhelming volume of data and high sequencing costs of metagenomic techniques. One common strategy is to combine amplicon sequencing with metagenomics: 16S rRNA amplicon sequencing technology is able to cope with a large sample size, enabling the selection of characteristic individuals for in-depth metagenomic analysis. Furthermore, the third-generation sequencing technology, which completes sequencing by reading entire DNA segments in a single pass, is vital for targeted sequencing and metagenomic sequencing of the gut microbiome of the red panda. This includes single-molecule real-time sequencing (SMRT), which uses DNA polymerase to read DNA in real time, and nanopore sequencing, which passes DNA through a tiny nanopore for sequencing. These faster and more accurate techniques are crucial for understanding the complex microbial communities in the red panda’s gut, surpassing the limitations of pyrophosphate sequencing technology. In addition, to classify microorganisms, a combination of this advanced sequencing technology and other methods, such as traditional cultivation techniques, should be employed.

Compared to the giant panda, our comprehension of the functional mechanisms that govern the impact of microorganisms in the red panda’s gut on its health and evolutionary adaptation is significantly lacking. Bridging this knowledge gap requires the integration of physiological, pathological and multi-omics analyses, including metagenomics, macrotranscriptomics, metaproteomics, metabolomics and more. Additionally, conducting comparative studies with other animals would enhance our insights into how the red panda adapts to its obligate bamboo diet through its gut microbiome.

Gut microbial symbiosis plays an important role in host immunity, nutrient utilization and disease and is mainly affected by food and nutrition composition [52,53,54]. Microorganisms in the bamboo phyllosphere may be an important influencing factor on the gut microbes of the red panda, including pathogenic microorganisms. Opportunistic pathogenic microorganisms in the gut of the red panda are the chief culprits in several gastrointestinal diseases, respiratory diseases, urinary system infections and circulatory system diseases in the red panda. Therefore, correlation studies should be conducted between these diseases and microorganisms such as *Klebsiella*, *Escherichia*, *Proteus*, *Acinetobacter* and other fungi, viruses and so on. Although a large number of studies have clarified the microbial composition, diversity and functions of the red panda gut, further studies of the different intestinal microbiome types (including the birth microbiome, captivity microbiome and microbiome after release from captivity) are required. This will help us protect the small populations of red pandas that are in danger of dying out. The gut microbiota of captive species could approach that of wild populations after being subjected to wild training, and some strains contributed to host dietary adaptation in the wild after release from captivity [55]. This would further aid in the conservation and restoration of endangered red panda populations in their natural environment.

Although current methods for detecting gut microbes in red pandas are constantly being updated, there are still many unanswered questions that require further investigation, such as the pathogenesis of gut microbes in canine distemper, the bacteria directly associated with gut stress syndrome in red pandas, the metabolic pathways involved in gut microbes and so on. In the future, research on the gut microbes of the red panda will also be carried out in line with the model conservation of the giant panda and the golden monkey to elucidate the micro-mechanisms of red panda disease and morphological expression through higher precision sequencing technology and metabolic monitoring at the amino acid level.

## Figures and Tables

**Figure 1 microorganisms-12-00478-f001:**
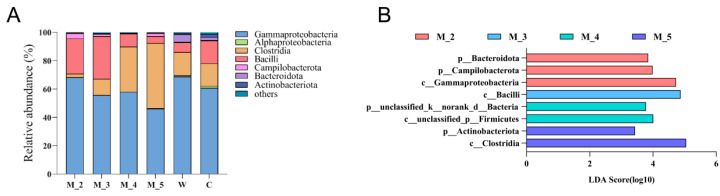
Relative abundance (**A**) and linear discriminant analysis (LDA) effect size (LEfSe) analysis (**B**) of the red panda cub gut microbiota for cubs of different ages at the phylum level [36].

**Figure 2 microorganisms-12-00478-f002:**
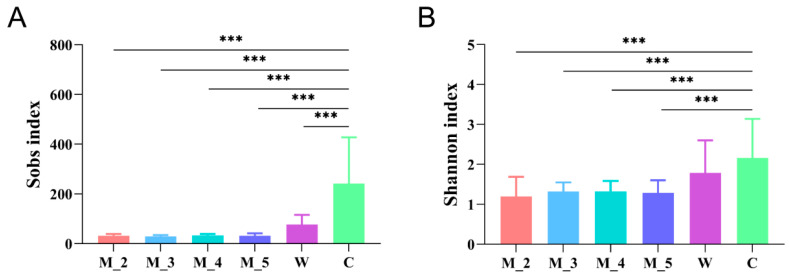
Differences in the Sobs (**A**) and Shannon (**B**) indices of the red panda cub gut microbiota for cubs of different ages at the OTUs level (Kruskal–Wallis H test, *** *p* ≤ 0.001) [36].

**Figure 3 microorganisms-12-00478-f003:**
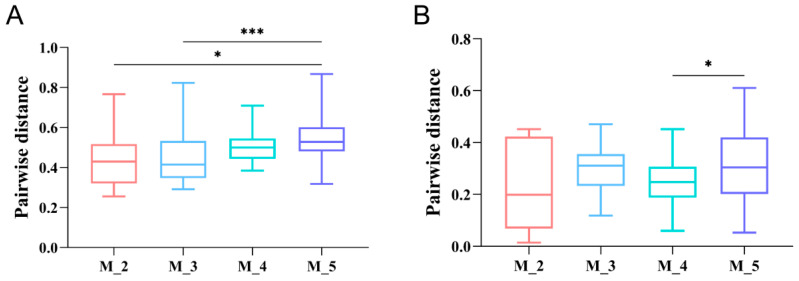
Inter-microbiome similarity of faecal samples from two- to five-month-old red panda cubs based on unweighted (**A**) and weighted (**B**) UniFrac distances (Kruskal–Wallis H test, * 0.01 < *p* < 0.05, *** *p* ≤ 0.001) [36].

**Table 1 microorganisms-12-00478-t001:** Methods for studying the gut microflora of the red panda.

Manufacturer/ Sequencing Platform	Objective Sequences	Captive/Wild	Sample Number: Captive/Wild	References
Roche/454 GS Junior	Hypervariable regions of the bacterial 16S rRNA gene	Captive	4	Crowe et al. (2013) [27]
		Captive	2	Williams et al. (2014) [28]
Roche/454 GS-FLX	V1–V3 hypervariable regions of the bacterial 16S rRNA gene	Captive/Wild	16/6	Kong et al. (2014) [29]
		Captive	6	Li et al. (2015) [31]
		Captive/Wild	4/4	Huang et al. (2020) [30]
Thermo Fisher/Ion PGM	V2, V3, V4, V6, V8 and V9 hypervariable regions of the bacterial 16S rRNA gene	Captive	2	McKenney et al. (2018) [32]
Illumina/MiSeq	V3–V4 hypervariable regions of the bacterial 16S rRNA gene	Captive	15	Williams et al. (2018) [33]
Illumina/Miseq PE300	V3–V4 hypervariable regions of the bacterial 16S rRNA gene	Captive	116	Long et al. (2022) [34]
		Captive/Wild	157/16	Wang (2023) [36]
	V1–V3 hypervariable regions of the bacterial 16S rRNA gene	Wild	103	Kang (2023) [35]
Illumina/HiSeq 2500	V4 hypervariable region of the bacterial 16S rRNA gene	Captive	1 (stomach, duodenum, jejunum, ileum, colon and rectum of one dead individual), 1 (faecal sample)	Zeng et al. (2018) [37]
Illumina/HiSeq 2000	Metagenome	Captive/Wild	4/4	Huang et al. (2020) [30]
Illumina/HiSeq 2500		Wild	6	Zhu et al. (2018) [18]
Illumina/Novaseq 6000		Wild	10	Kang (2023) [35]

## Data Availability

No new data were created or analyzed in this study. Data sharing is not applicable to this article.

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
