# Peer review of "Progress in Research on the Gut Microflora of the Red Panda (Ailurus fulgens)"

_microorganisms, 2024, doi:10.3390/microorganisms12030478_

Round 1

Reviewer 1 Report

Comments and Suggestions for Authors

This review provides valuable insights into the gut microbiota of red pandas and presents papers on the effects of age and habitat on microbial diversity. These papers utilize advanced sequencing techniques to identify the predominant bacterial groups and their role in health and nutrition. Differences in diversity indices among various age groups and between captive and wild pandas highlight the importance of the gut microbiome in red panda development and health, but may lead to different conclusions in different papers. This review attempts to bring these papers together and compare them in order to provide a reference for the conservation of the endangered red panda. Overall, the review is clearly organized and interesting to read. I would like the authors to consider revising only the following point.

Major point

The data shown by Figure 2 appear to be the results of the authors' paper, but the conclusions are opposite to the results of other studies in terms of the high diversity of the bacterial flora of captive red pandas. This may be misleading. The review should present a balanced overview, incorporating not only the authors' results but also findings from various disciplines. To address this point, I suggest a comparison with the literature to identify the reasons for these discrepancies and to ensure comprehensive and fair representation in the review.

Author Response

Dear Editors and Reviewers:

Thank you for your and reviewers’ comments concerning our manuscript (ID, microorganisms-2885856) entitled “Progress in research on the gut microflora of the red panda (Ailurus fulgens)”. Those comments are all valuable and very helpful for revising and improving our paper, as well as the important guiding significance to our researches. We have studied comments carefully and have made correction which we hope meet with approval. Revised portion are marked in yellow in the revised-manuscript. The main corrections in the paper and the responds to the reviewers’ comments are as flowing:

Reviewer 1:

(1) The data shown by Figure 2 appear to be the results of the authors' paper, but the conclusions are opposite to the results of other studies in terms of the high diversity of the bacterial flora of captive red pandas. This may be misleading. The review should present a balanced overview, incorporating not only the authors' results but also findings from various disciplines. To address this point, I suggest a comparison with the literature to identify the reasons for these discrepancies and to ensure comprehensive and fair representation in the review.

Re: Thanks to the constructive comments of the expert, we have synthesised the results of previous studies and our findings in Figure 2 for careful comparison, and have explained the reasons for discrepancies between our results and those of other studies to ensure a full and fair presentation in the review. L191-192, L199-201 in revised-manuscript.

Reviewer 2 Report

Comments and Suggestions for Authors

In a few sentences, there are complex structures that might be simplified for better readability. For example, consider breaking down sentences like "Using the Illumina/ Miseq PE300 platform targeting the V1-V3 and V3-V4 hypervariable regions of the bacterial 16S rRNA gene, respectively, we found significant differences in diversity between the different seasons [34,35], and these were significantly influenced by growth and development [36]."

Some sentences present findings but might benefit from a brief explanation or clarification. For instance, after mentioning "significant differences in diversity between the different seasons [34,35]," you may want to briefly explain what these differences are or why they are significant.

The tense used is generally in the past tense, which is appropriate for describing research methods and findings. However, ensure consistency in the tense throughout the text.

Summary and prospects

While the content is comprehensive and informative, consider presenting key concepts more succinctly. Use shorter sentences and paragraphs to facilitate easier comprehension for the reader.

Provide a bit more detail on modern technologies such as single-molecule real-time sequencing and nanopore sequencing. The reader may benefit from a brief explanation of why these technologies are essential for studying the red panda's gut microbiome.

Explain how third-generation sequencing, such as single-molecule real-time sequencing and nanopore sequencing, contributes to a better understanding of the red panda's gut microbiome.

Emphasize the latest and most significant discoveries, especially those that may impact the conservation and restoration of endangered red panda populations in their natural environment.

Add a few words about what questions or research directions remain open and require further investigation to better understand the red panda's gut microbiome.

Author Response

Dear Editors and Reviewers:

Thank you for your and reviewers’ comments concerning our manuscript (ID, microorganisms-2885856) entitled “Progress in research on the gut microflora of the red panda (Ailurus fulgens)”. Those comments are all valuable and very helpful for revising and improving our paper, as well as the important guiding significance to our researches. We have studied comments carefully and have made correction which we hope meet with approval. Revised portion are marked in yellow in the revised-manuscript. The main corrections in the paper and the responds to the reviewers’ comments are as flowing:

Reviewer 2:

  •  In a few sentences, there are complex structures that might be simplified for better readability. For example, consider breaking down sentences like "Using the Illumina/ Miseq PE300 platform targeting the V1-V3 and V3-V4 hypervariable regions of the bacterial 16S rRNA gene, respectively, we found significant differences in diversity between the different seasons [34,35], and these were significantly influenced by growth and development [36]."

Re: Thanks to the comments of the expert, We have broken down sentences with complex structures in the text in the hope of improving the flow and readability of the sentences. L107-111 in revised-manuscript.

  • Some sentences present findings but might benefit from a brief explanation or clarification. For instance, after mentioning "significant differences in diversity between the different seasons [34,35]," you may want to briefly explain what these differences are or why they are significant.

Re: Thank you very much for the constructive suggestions given by the experts, We have broken down sentences with complex structures in the text in the hope of improving the flow and readability of the sentences. L109-111 in revised-manuscript.

  • The tense used is generally in the past tense, which is appropriate for describing research methods and findings. However, ensure consistency in the tense throughout the text.

Re: Thanks to the expert's question about the tense of the article, the description of the chronological development of the text, there are things that will be mentioned in the recent progress of research, which led to our error in the tense, we have carefully checked and corrected the grammatical problems of the whole text. L40, L50, L53, L88, L115, L119,L122, L125-126, L143,L145, L147, L150, L152-153, L158-159, L185-186, L211, L227, L262, L264-265, L271, L275, L277, L282-284, L286-287, L290, L293, L296, L298-299, L305, L307, L309-310, L312, L315-316, L363 in revised-manuscript.

Summary and prospects

  • While the content is comprehensive and informative, consider presenting key concepts more succinctly. Use shorter sentences and paragraphs to facilitate easier comprehension for the reader.

Re: Thanks to the experts' suggestions on the reading experience, we have re-split the long difficult sentences in the Summary and Outlook section, in the hope that readers will be able to understand the past and future of this type of research more easily in a limited space. L319, L329-330, L362-364 in revised-manuscript.

  • Provide a bit more detail on modern technologies such as single-molecule real-time sequencing and nanopore sequencing. The reader may benefit from a brief explanation of why these technologies are essential for studying the red panda's gut microbiome.

Re: Appreciate the input of experts to improve the reader's experience, and we have used short, easy-to-read sentences in the appropriate paragraphs to explain single-molecule real-time sequencing and nanopore sequencing technologies, which we hope will be the easiest for readers to understand without adding to their reading load. L334-339 in revised-manuscript.

  • Explain how third-generation sequencing, such as single-molecule real-time sequencing and nanopore sequencing, contributes to a better understanding of the red panda's gut microbiome.

Re: Thanks to the experts' comments, We have added a few short sentences to explain the advantages of third-generation sequencing and the breakthroughs it will bring to the study of gut microbiome in red pandas. L339-341 in revised-manuscript.

  • Emphasize the latest and most significant discoveries, especially those that may impact the conservation and restoration of endangered red panda populations in their natural environment.

Re: Thanks to expert advice on the completeness of the article, we have added a few sentences to highlight recent research findings that are important for the conservation and recovery of endangered wild red panda populations. L365-369 in revised-manuscript.

  • Add a few words about what questions or research directions remain open and require further investigation to better understand the red panda's gut microbiome.

Re: Thanks to expert advice for the completeness of this review, we have added a section to discuss the questions that remain unanswered in current research on the red panda and to provide an outlook on future directions for red panda gut microbiology. L370-378 in revised-manuscript.

Round 2

Reviewer 1 Report

Comments and Suggestions for Authors

I support the acceptance of the revised version, taking into account the authors' opinions.